# Electrochemical Detection of Electrolytes Using a Solid-State Ion-Selective Electrode of Single-Piece Type Membrane

**DOI:** 10.3390/bios11040109

**Published:** 2021-04-07

**Authors:** Li-Da Chen, Wei-Jhen Wang, Gou-Jen Wang

**Affiliations:** 1Department of Mechanical Engineering, National Chung-Hsing University, Taichung 40227, Taiwan; 40332131@gm.nfu.edu.tw (L.-D.C.); freedom84528@gmail.com (W.-J.W.); 2Graduate Institute of Biomedical Engineering, National Chung-Hsing University, Taichung 40227, Taiwan; 3Regenerative Medicine and Cell Therapy Research Center, Kaohsiung Medical University, Kaohsiung 80708, Taiwan

**Keywords:** solid-state ion-selective electrode, electrochemical detection of electrolytes, cyclic voltammetry, square-wave voltammetry

## Abstract

This study aimed to develop simple electrochemical electrodes for the fast detection of chloride, sodium and potassium ions in human serum. A flat thin-film gold electrode was used as the detection electrode for chloride ions; a single-piece type membrane based solid-state ion-selective electrode (ISE), which was formed by covering a flat thin-film gold electrode with a mixture of 7,7,8,8-tetracyanoquinodimethane (TCNQ) and ion-selective membrane (ISM), was developed for sodium and potassium ions detection. Through cyclic voltammetry (CV) and square-wave voltammetry (SWV), the detection data can be obtained within two minutes. The linear detection ranges in the standard samples of chloride, sodium, and potassium ions were 25–200 mM, 50–200 mM, and 2–10 mM, with the average relative standard deviation (RSD) of 0.79%, 1.65%, and 0.47% and the average recovery rates of 101%, 100% and 96%, respectively. Interference experiments with Na^+^, K^+^, Cl^−^, Ca^2+^, and Mg^2+^ ions demonstrated that the proposed detection electrodes have good selectivity. Moreover, the proposed detection electrodes have characteristics such as the ability to be prepared under relatively simple process conditions, excellent detection sensitivity, and low RSD, and the detection linear range is suitable for the Cl^−^, Na^+^ and K^+^ concentrations in human serum.

## 1. Introduction

The chloride, sodium and potassium ions in the human body can maintain the physiological functions of water metabolism, the neuromuscular system, acid-base balance and osmotic pressure in the body. Therefore, the metabolic functions of the human body can be evaluated based on the measurements of the chloride, sodium and potassium ions concentrations in the body [1,2,3,4].

The concentrations of chloride, sodium, and potassium ions in normal human serum are 98–106 mM [2], 135–145 mM [3] and 3.5–5 mM [3], respectively. Too-high concentrations in the human body will cause hyperchloremia, hypernatremia, and hyperkalemia, while the occurrence of concentrations lower than the normal concentration will cause hypochloremia, hyponatremia, and hypoxemia potassium. Another common chloride-related disease is cystic fibrosis, which is an autosomal recessive genetic disease that affects mucus-producing cells in the body and causes mucus accumulation in the lungs, trachea, and digestive tract, leading to the degradation of breathing functions [5]. The general clinical diagnosis of this disease is based on measuring the chloride ion content in sweat. People with the disease usually have a chloride ion content of over 60 mM in their sweat [2], while the chloride ion content in normal human sweat is below 40 mM [5].

The laboratory autoanalyzer is a general electrolyte-detecting instrument that can provide doctors with reliable results. However, the specimen must be delivered to a specific laboratory and diluted before indirect ion-selective electrodes (ISEs) are used for detection. For patients with a severe or life-threatening degree of the illness, the blood gas analyzer (BGA) is usually used clinically to achieve rapid detection. The BGA uses direct ISE for direct analysis without the need to first dilute the sample, and the results can be obtained within 1 to 5 min [6,7,8,9,10]. The BGA can be divided into two types: benchtop and portable. The benchtop BGA detection cartridge has a service life of about 28–30 days; hence, it is suitable for mass detection applications. However, the instrument needs regular calibration and maintenance. The portable BGA usually uses a disposable detection cassette, which can reduce the frequency of instrument calibration and maintenance; however, since the cassette cannot be reused, the portable BGA is only suitable for small-volume detection applications [11]. In recent years, as the utilization rate of point-of-care (POC) testing has significantly increased, the economic value of POC-related products has also increased [12]. Therefore, there are many portable and small BGAs and electrolyte analyzer-related products in the market, such as Abbott i-STAT [13], Siemens epoc^®^ blood analysis system [12], and Arkray Spotchem EL SE-1520 [14].

At present, the electrochemical potentiometric method [13,14,15,16,17,18] is the main method for electrolyte ions quantification. Other detection methods include the conductivity method [19,20], cyclic voltammetry (CV) [21,22,23,24,25,26], and coulometry [27,28]. The potentiometric method uses liquid-state ISEs that need to be regularly replaced. The operating temperature is limited, and the electrodes cannot be effectively miniaturized [29,30,31]. Moreover, the practical application of the conductance method is not yet mature [21]. Therefore, in the current study, a novel detection method combining high-sensitivity voltammetry and solid-state ISE is developed. The solid-state ISE does not need to be filled with a solution. The ion-selective membrane directly contacts the electrode, and the solid contact material of the intermediate layer is added to enhance the effect of ion and electron conversion. Therefore, the sensor can be more easily miniaturized, and its durability and operating temperature range are improved.

Commonly used solid contact materials include high-molecular-weight conductive polymers such as polypyrroles [32], poly(3-octylthiophene) [33], and poly(3,4-ethylenedioxythiophene) (PEDOT) [34]. Nanomaterials used as solid contact materials include carbon nanotubes [17], graphene [35], platinum nanoparticles [36], and noble metal nanostructures [37]. Electroactive substances are also used, such as tetrathiafulvalene [38] and 7,7,8,8-tetracyanoquinodimethane (TCNQ) [39]. Conductive polymers such as PEDOT need to be synthesized via electrochemical polymerization [34], while nanomaterials need to be produced via electrochemical deposition [37]. The manufacturing processes of both are relatively cumbersome. Therefore, this study uses the solid-state contact conduction mechanism of high redox capacitors [29] and adopts the electroactive material TCNQ as the solid-state contact material.

TCNQ is a powerful organic electron acceptor with excellent electrical properties and high electron affinity (2.88 eV) [40]; thus, it can easily donate electron acceptors to react with transition metals or organic compounds and form numerous TCNQ anion–based salt complexes, such as CuTCNQ [40], AgTCNQ [40], KTCNQ [39], and NaTCNQ [39]. In addition, the use of the TCNQ involves a simple process; the TCNQ only needs to be dropped on the electrode substrate through drop-casting [39].

In this study, electrodes for the effective and fast detection of electrolytes in human serum were developed. A flat thin-film gold electrode was used as the detection electrode for chloride ions and a solid-state ISE, which was formed by covering a flat thin-film gold electrode with a mixture of TCNQ and ion-selective membrane (ISM), was developed for sodium and potassium ions detection, respectively.

## 2. Materials and Methods

### 2.1. Materials and Reagents

TCNQ, tetrahydrofuran (THF), potassium chloride, sodium chloride, calcium chloride, magnesium chloride, human serum, potassium ionophore I, poly(vinyl chloride) and potassium phosphate monobasic were purchased from Sigma-Aldrich (St. Louis, MO, USA). Potassium ferricyanide and potassium ferrocyanide were purchased from SHOWA (Tokyo, Japan). Potassium tetrakis (4-chlorophenyl)borate, 2-nitrophenyl octyl ether, and 4-tert-butylcalix[4]arenetetraacetic acid tetraethyl ester were purchased from Thermo (Darmstadt, Germany). Sodium phosphate dibasic was purchased from J. T. Baker (USA). Acetone was purchased from ECHO Chemical (Miaoli, Taiwan). Distilled deionized water (>18 MΩ) was obtained from ELGA LabWater (London, UK).

### 2.2. Electrode Fabrication

Scheme 1 illustrates the fabrication process of the chloride, potassium, and sodium ions detection electrodes. First, a thin gold layer was sputtered onto a 3 × 5 mm^2^ polyethylene terephthalate (PET) membrane (Scheme 1A). Then, the gold thin-film–sputtered PET membrane was packaged to form the detection chip (Scheme 1B). For the potassium and sodium ions detection electrodes, 3 μL of a TCNQ and ISM mixture solution was dripped onto the electrode (Scheme 1C). Table 1 presents the compositions and contents of potassium and sodium ISMs.

### 2.3. Apparatus

Cyclic voltammetry and square-wave voltammetry (SWV) were conducted with an SP-150 potentiostat (BioLogic, Seyssinet-Pariset, France) connected to a computer. A three-electrode system, consisting of a working electrode, a platinum counter electrode, and an Ag/AgCl reference electrode, was applied. Field-emission scanning electron microscopy (FE-SEM, Zeiss UltraPlus, Potsdam, Germany) was used for the electrode surface characterization. The analysis system and the electrodes are shown in Figure 1.

## 3. Results and Discussion

### 3.1. Optimization of Detection Electrodes

Two parameters of the detection electrode need to be optimized—the organic solvent to dissolve TCNQ and the volume ratio of TCNQ to ISM. Regarding the organic solvent, a suitable solvent for TCNQ was selected from two organic solvents: acetone and THF. The TCNQ solution was mixed with the corresponding ISM (Na^+^ and K^+^) with a volume ratio of 1:1 to determine which was the more suitable solvent. After the suitable TCNQ solvent was selected, TCNQ/ISM mixed solutions with different volume ratios (1:1, 1:2, 2:1, 3:1) were prepared to optimize the volume ratio.

Figure 2A,B show the cyclic voltammograms for Na^+^ (140 mM NaCl) and K^+^ (5 mM KCl) detections, respectively. The acetone-dissolved TCNQ resulted in peak oxidation currents at 0.46 and 0.50 V for Na^+^ and K^+^ detections, respectively. The peak oxidation currents were much higher than those obtained from the THF-dissolved TCNQ. Therefore, acetone was selected as the TCNQ solvent. The cyclic voltammograms for Na^+^ and K^+^ detections with different TCNQ/ISM volume ratios are shown in Figure 2C,D, respectively. The results show that for the Na^+^ detection, the TCNQ/ISM volume ratio of 2:1 resulted in the maximum peak oxidation current at 0.45 V, while for the K^+^ detection, the ratio of 3:1 resulted in the maximum peak oxidation current at 0.43 V. In Figure 2C,D, as the TCNQ content increases, the oxidation potentials of Na^+^ and K^+^ shift to the left. Therefore, using a lower potential can drive the oxidation reaction and reduce the scanning voltage range during CV.

### 3.2. Characterization of Fabricated Electrodes

Figure 3 displays the scanning electron microscopy (SEM) images of the fabricated electrodes. The top view shows many typical TCNQ black blocks and a few white flake structures. The white flake structures are TCNQ erected under the influence of gravity [41]. The cross-sectional view (Figure 3B) shows that the thickness of the deposited Na^+^ ISM and K^+^ ISM layers: about 24.4 and 14.6 µm, respectively.

### 3.3. The Electrochemical Behavior of Cl^−^, Na^+^, and K^+^ Electrodes

The electron diffusion rate can be used to study the characteristics of electrochemical electrodes. The oxidation and reduction potentials, as well as the reaction current of the analyte, can be obtained based on a CV analysis of the reaction between the analyte and the electrode. The results can be used to determine whether the reaction is diffusion-controlled.

Figure 4 shows the CV results of the proposed different electrodes at scan rates ranging from 100 to 500 mV/s in 50 mM KCl, 140 mM NaCl, and 1 mM KCl solutions. The cyclic voltammograms displayed in Figure 4(A1–C1) show that the absolute value of the oxidation and reduction peak currents increased with the increase in scan rate. Generally, the Randles–Sevcik equation (Equation (1) is used to evaluate the diffusion-controlled characteristic of a sensing electrode.
(1)$$i_{p}=2.69\times 10^{5}\times n^{3/2}\times A\times C\times D^{1/2}\times v^{1/2}$$
where *i_p_* is the peak current, *n* is the number of electrons appearing in a half-reaction for the redox couple, *A* is the electrode area (cm^2^), *C* is the analyte concentration (mole/cm^3^), *D* is the analyte diffusivity (cm^2^/s), and *v* is the scan rate of potential (V/s). For stationary *n*, *A*, *C*, and *D*, *i_p_* is a linear function of the square root of the scanning rate. The corresponding linear relationship between the peak current and the square root of the scanning rate of the proposed electrodes for the cyclic voltammograms shown in Figure 4A1–C1 are given in Figure 4A2–C2, respectively. The *R*^2^ values were calculated to be 0.9986, 0.9981, and 0.9989, respectively. The results demonstrate that the proposed electrodes possess a typical diffusion-controlled electrochemical characteristic; hence, the electrodes are suitable for practical quantitative analyses.

### 3.4. Detection of Cl^−^, Na^+^, and K^+^

Cyclic voltammetry was used for the detection of Cl^−^ (KCl) because it can be easily oxidized by gold with a suitable oxidation potential. Square-wave voltammetry was adopted for the detections of Na^+^ (NaCl) and K^+^ (KCl) because the oxidation process involved the ISM.

(1)Cl^−^ detection

Figure 5 shows the detection results of Cl^−^ with various concentrations (25, 50, 100, 150, and 200 mM). Clear oxidation and reduction current peaks can be observed in Figure 5A. The absolute value of the oxidation and reduction peak currents increased with the increase in Cl^−^ concentration. Figure 5B displays the corresponding calibration curve for Cl^−^ detection, with the normal concentration range of Cl^−^ in human blood identified by a bar icon. The linear detection range, sensitivity, correlation coefficient, relative standard deviation (RSD), and recovery rate of the electrode were calculated to be 5–200 mM, 0.27 mA mM^−1^ cm^−2^, 0.9988, 0.79%, and 101%, respectively.

(2)Na^+^ detection

Figure 6 shows the SWV detection results of Na^+^ with various concentrations (50, 100, 150, and 200 mM). Figure 6A displays clear oxidation current peaks, which increase with the increase in the Na^+^ concentrations. Figure 6B shows the corresponding calibration curve for Na^+^ detection, with the normal Na^+^ concentration range in human blood marked by a bar icon. The linear detection range, sensitivity, correlation coefficient, RSD, and recovery rate of the electrode were calculated to be 50–200 mM, 0.125 μA·mM^−1^cm^−2^, 0.9786, 1.65% and 100%, respectively.

(3)K^+^ detection

The SWV detection results of K^+^ with various concentrations (2, 3.5, 5, 7.5, and 10 mM) are shown in Figure 7. In Figure 7A, the oxidation current peaks increase with increasing K^+^ concentrations. Figure 7B shows the corresponding quadratic calibration curve for K^+^ detection, which is more appropriate for fitting the measured data. The detection range, sensitivity, correlation coefficient, RSD, and recovery rate of the electrode were measured to be 2–10 mM, 0.56–7.98 μA·mM^−1^ cm^−2^, 0.9778, 0.47%, and 96%, respectively.

(4)Interference detection

We used normal concentrations of Na^+^, K^+^, Cl^−^, Ca^2+^, and Mg^2+^ in human blood to examine the influence of interference substances on the oxidation current response of different electrodes. The experimental results are depicted in Figure 8.

The results of the Cl^−^ detection interference experiments (Figure 8) showed that the oxidation currents of four chlorides (KCl, NaCl, CaCl_2_, and MgCl_2_) having the same concentration (100 mM) were between 1.84 and 1.94 mA, with an average current of 1.89 mA. The RSD was calculated to be 2.35%. The experimental results indicate that the Cl^−^ detection electrode was not interfered by cations.

The results of the Na^+^ detection interference experiments (Figure 8) showed that the current response of the Na^+^ detection electrode in the NaCl solution was 5.55 µA. After KCl, CaCl_2_, and MgCl_2_ were added to the KCl solution, the current interferences were measured to be 1.08%, 7.31%, and 1.68%, respectively. Among the introduced compounds, CaCl_2_ had the greatest impact on the current response. However, there was no statistically significant difference between the current response of the NaCl solution containing any of the three interference substances and that of the pure NaCl solution. The experimental results show that the proposed Na^+^ detection electrode has good selectivity.

As shown in Figure 8, the current response of the K^+^ detection electrode in pure KCl solution was measured to be 8.92 µA. The current interferences after the addition of NaCl, CaCl_2_, and MgCl_2_ were measured to be 5.23%, 1.01%, and 4.21%, respectively. Similar to the results of the interference experiment of the Na^+^ detection electrode, there was no statistically significant difference between the current response of the KCl solution containing any of the three interference substances and that of the pure KCl solution. The experimental results show that the proposed K^+^ detection electrode possesses good selectivity.

### 3.5. Performance Comparisons

In recent years, the electrochemical techniques for measuring electrolyte concentration have been rapidly increasing. The materials for Cl^−^ detection electrode are mostly nano- and micro-structures of carbon, silver, and platinum. Sodium-ion detection electrode materials are mainly manganese oxide–modified carbon and iron sulfate–modified glass carbon. Potassium ion electrode materials include manganese oxide nanorod–modified carbon electrode as well as carrier- and DNA-modified gold electrode. Table 2 compares the detection performances of the proposed detection electrodes and the existing technologies. The proposed detection electrode has characteristics such as the ability to be prepared under relatively simple process conditions, excellent detection sensitivity, and low RSD, and the detection linear range is suitable for the chloride, sodium, and potassium ions concentrations in human serum.

## 4. Conclusions

In this study, Cl^−^, Na^+^, and K^+^ detection electrodes were successfully developed through a simple process to effectively improve the electrolyte detection speed and obtain an appropriate linear detection range, good sensitivity, RSD and recovery rate. Through CV and SWV, the detection data can be obtained in two minutes. The linear detection ranges in the standard samples of Cl^−^, Na^+^, and K^+^ were measured to be 25–200 mM, 50–200 mM, and 2–10 mM, with the average RSDs of 0.79%, 1.65%, and 0.47%, respectively, and the average recovery rates of 101%, 100% and 96%, respectively. The proposed detection electrode has characteristics such the ability to be prepared under relatively simple process conditions, excellent detection sensitivity, and low RSD; moreover, the detection linear range is suitable for the Cl^−^, Na^+^, and K^+^ concentrations in human serum.

## Data Availability

The data that support the findings of this study are available from the corresponding author upon reasonable request.

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
