# Peer review of "Electrochemical Detection of Electrolytes Using a Solid-State Ion-Selective Electrode of Single-Piece Type Membrane"

_biosensors, 2021, doi:10.3390/bios11040109_

Round 1
Reviewer 1 Report
The authors developed simple electrochemical electrodes and demonstrated the fast detection of chloride, sodium, and potassium ions. The linear detection ranges in the standard samples of chlorine, sodium, and potassium ions were 25–200 mM, 50–200 mM, and 2–10 mM, with the average relative standard deviation (RSD) of 0.79%, 1.65%, and 0.47% and the average recovery rates of 101%, 100%, and 96% respectively. Interference experiments with Na+, K+, Cl−, Ca2+, and Mg2+ ions demonstrated that the proposed detection electrodes have good selectivity.
The figures are also very clear and comprehensible. I will be happy to recommend this work for publication once the following points have been addressed.
- In the abstract, the authors mentioned developed electrodes for the fast detection of ions in human serum. It may be due to their linear detection ranges but the authors have not tested their electrodes on actual human serum and it can mislead readers. Please provide more clarification.
- The authors used two interchangeable terms, chlorine ion and chloride ion, but I believe the authors need to use only one term through the manuscript for consistency.
- There are some typos and citation format is not consistent. Please look line 31, 64, 102, 106, and 151
- The authors synthesized a mixture of TCNQ and ion-selective membrane and developed solid-state ISEs to detect sodium and potassium ions. However, it has been shown by other groups (reference #1 below). They used carbon disc-TCNQ-sodium or potassium ISM. What is the novelty of your work? Perhaps different manufacturing process, improved electrode stability and reliability?
- The authors mentioned liquid-state ISEs have limited operating temperature while solid-state ISEs show improved operating temperature range. What was testing temperature? Have you tested the electrodes in different temperatures to see the effect of temperature on your electrode performance?
- The authors described sensor sensitivity and selectivity in the text, but not its durability and stability. How many times are the electrodes reusable? Have you tested any potential drift of the electrodes for the long period of time (e.g., a day or a week)?
- Paczosa-Bator, M. Pięk, and R. Piech, Application of nanostructured TCNQ to potentiometric ion-selective K and Na electrodes, Analytical Chemistry, 2015, 87, 1718-1725.
Reviewer 2 Report
This manuscript reports a simple electrochemical detection of the chlorine, sodium, and potassium using the TCNQ/ISM membrane. Overall quality of this work is satisfactory, but requires few supporting materials and additional clarification on the different locations. I recommended to publish this manuscript with “minor revision”. The following comments will be helpful to improve the overall quality of this manuscript.
Comments:
- Authors show the schematics for the electrode fabrication. However, it also needs to provide the real images, including the electrodes and analysis system in order to enhance the readers’ understanding.
- Why did the author comparison with only two organics of acetone and THF among the various solvents? The experimental results show the organic solvents is critical factors for the sensitivity even though the same TCNQ/ISM material is used. Also, please provide why the solvents used in the deposition make the difference for the sensor characteristics.
- Authors show the top SEM view for the TCNQ/ISM-deposited electrode, however, the cross-sectional SEM view indicates the Na+ISM and K+ISM. What is the Na+ISM and K+ISM?
- Please confirm the units or the references on the following sentences: line 7, page 1; line 18, page 2; and line 7, page 3.
